
# A scheme to detect sand/dust weather applying meteorological radars

Xuebang Gao, Li Xie

College of Civil Engineering and Mechanics, Lanzhou University, Lanzhou, Gansu 730000, China;Key Laboratory of Mechanics on Disaster and Environments in Western China, Attached to Ministry of Education, Lanzhou University, Lanzhou, Gansu 730000, China

*Correspondence to*: Li Xie (xieli@lzu.edu.cn)

**Abstract** Sandy dust weather occur frequently in arid and semi-arid areas. It is important to actually detect the sandy dust grain concentration or the visibility of the sandy dust weather for weather forecasting. In this paper, based on numerical calculation of the effective detection distance of different radar detecting the sandy-dust weather with different strength, a

scheme to detect sand/dust weather applying existed meteorological radar stations is proposed in this paper. The scheme can be efficient to detect sandy dust weather, for it makes a good supplement to the current deficiencies in detecting sandy dust weather and it's a cost-saving detection way by using the existed meteorological radars. In addition, the effect of charges carried by sand/dust grains and the relative humidity on the effective detection distance of radar is also investigated, and it shows that these effects will not change the proposed scheme. It will be promising to detect the sandy dust weather in the way

of disastrous weather precaution by using this scheme.

## 1 Introduction

Sandy dust weather are common disastrous weather in many parts of the world (Cao et al., 2015; Zhao et al., 2015; Al-Hemoud et al., 2020; Lee et al., 2009; Middleton, 2017). Strong sandy dust weather can cause traffic accidents and communication signal interruption, etc(Baddock et al., 2013). For example, in March 2021, the severe sandstorm swept

Mongolia, northern China and other parts of Asia, causing dozens of deaths, missing livestock, continuous disruptions of communications and power, and the sand and dust weather affected an area of more than 3.8 million square kilometres (WMO, 2021). Therefore, the timely detection of sand and dust weather and its intensity is not only the basis of issuing accurate sand and dust weather warning, but also provides a guarantee for reducing the harm of sand and dust storms to the maximum extent(Al-Hemoud et al., 2017; Akhlaq et al., 2012).

In the field of detecting sand and dust weather, meteorological satellite passive remote sensing and laser radar (lidar) active remote sensing are commonly used techniques to detect sand and dust weather(Akhlaq et al., 2012; Ma et al., 2011; Gao et al., 2017; Pan et al., 2020; Miller et al., 2019). The coverage of meteorological satellite remote sensing is wide, and it has gradually become an effective means to monitor the characteristics of dust aerosols around the world, what's more, passive satellite remote sensing can effectively monitor the coverage and transport path of sandstorms(Akhlaq et al., 2012). However, the

strength distribution characteristics of each height layer of strong sand and dust weather cannot be obtained quantitatively(Chen



et al., 2010). In fact, the lidar can detect the vertical concentration distribution of sandy dust weather, but in the sandy dust weather with high concentration, the lidar wave will have serious signal attenuation(Akhlaq et al., 2012), resulting in the final identification and classification error or cannot receive a valid signal. For example, in the recent study by Pan et al.(Pan et al., 2020), in sandstorm weather, lidar misjudged dust as cloud at the height of 4km above the ground also with a serious signal attenuation. In addition, Chen et al detected the dust layer through lidar and infrared pathfinder satellite and found that about 43% of the dust layer (mainly dense dust layer) in the Taklimakan Desert was mistaken for clouds(Chen et al., 2010). It can be seen that serious misjudgements will occur when lidar detects strong sandy dust weather. On the other hand, as we all know, microwave radar is widely distributed in monitoring stations all over the world, and it is an important means of meteorological monitoring, and plays a very important role in sudden and disastrous monitoring, forecasting and alarm. Recent field measurements have found that effective echoes have been received when using microwave radar to measure sandstorms, which provides an idea for using microwave radar to detect sandy dust weather(Wang et al., 2017; Ming et al., 2019). In addition, some studies have found that dust particles carry a large amount of net charge(Zheng et al., 2003; Bo et al., 2014; Zhang and Zhou, 2020), and the charge carried by particles will enhance the backscattering of radar waves and the attenuation of radar waves on the back-and-forth path.(Qin and Li, 2015; Kocifaj et al., 2021). At the same time, when observing in the field, the researchers also found that the transit of sandstorm will cause the phenomenon of moisture inversion(Elsheikh et al., 2017), and the study shows that the relative humidity of the environment can also affect the scattering and absorption of electromagnetic waves by particles(Dou and Xie, 2017).

Therefore, considering the net charge carried by dust particles and the inverse humidity phenomenon of sand and dust weather, the scattering characteristics of sand and dust particles to electromagnetic waves are fully studied, and calculate the echo power of different wave band radar in sandy dust weather with different strength. Based on the high spatial resolution of laser radar and the low attenuation characteristics of microwave radar, the radar of different bands can be combined to detect sandy dust weather and its strength. In this paper, firstly, the scene of sand and dust weather is simulated, and the effective detection range of microwave radar and lidar for measuring sandy dust weather with different strength is calculated theoretically. Based on the effective detection range of radar in each band in detecting sandy dust weather with different strength, a scheme of comprehensive detection of sandy dust weather strength by combining different band radars is put forward. In addition, we also studied the effects of relative humidity and particle charge on radar echo power, so that the strength information of sandy dust weather can be detected more accurately.

## 2 Meteorological radar detects sandy dust weather

### 2.1 Sandy Dust Weather Model

Sandy dust weather is a kind of disastrous weather that occurs in arid and semi-arid areas (especially in the arid regions of the Middle East and Asia). It can be divided into four categories: floating dust with the visibility $V<10$km, blowing sand 1 km$<V<10$km), sandstorms $V<1$ km and severe sandstorms V<0.5 km. The strong sandy dust weather covers an area of several



hundred square kilometers, or affects an area within 10 km. Meteorological radars are usually used to detect the sandy dust weather. The sandy dust weather is classified by the radar detection distance. This paper mainly concerned about the applicable

range of meteorological radar with different wave bands to detect sandy dust weather with different strength. In order to facilitate the calculation, we simplify the sandy dust weather model, without considering the change of the concentration along the sand and dust weather horizontal transport path, and along the vertical height $h$ above the ground, the concentration of sandy dust particles, $N_h$, still is considered decreasing with height refer to (Gillette et al., 1974):

$$N_h = N_0 h^{-\Gamma p} \, ,$$   (1)

In which $N_0$ (1/m$^3$) is the number of sand and dust particles per unit volcume (particle concentration) at the height of 1m from the ground; $\Gamma_p$ is a coefficient related to particle sedimentation rate and friction velocity, $\Gamma_p = 0.29$ (Gillette et al., 1974).The visibility $V$ (in km) of sandy dust weather can be calculated by the particle concentration $N_0$ according to Goldhirsh's formula (Goldhirsh, 2001), $V = \dfrac{5.5 \times 10^{-4}}{N_0 a_e^2}$ , where $a_e$ is the equivalent radius of the particle system, which is calculated by $a_e = \left[ \int p(r) r^3 dr \right]^{1/3}$ given the particle size distribution density $p(r)$. Generally, particles with a diameter greater than 80

microns are difficult to be directly blown to a height of more than 2m by the wind (Abdulwaheed et al., 2014), while smaller dust particles can rise to a height of several kilometers by strong wind, and can be transported to a long distance. Therefore, the sandy dust particles less than 80 microns are chosen and the particle size distribution obeys lognormal distribution as follows (Ahmed et al., 1987)

$$p(r) = \frac{1}{\sqrt{2\pi} \ln \sigma_r} \exp\left[ -\frac{\ln r - \ln \bar{r}}{2(\ln \sigma_r)^2} \right] ,$$   (2)

Where $\bar{r}$ and $\sigma_r$ are the average and standard deviation of the particle radius, respectively.

## 2.2 Calculation of Radar Echo Power through Sandy Dust Weather

When radar waves go through sandy dust weather, due to scattering by dust particles, some radar signals will be returned to a receiver. By using the power of the return radar signals (radar echo power), the particle concentration can be reversely calculated, therefore it is important to obtain the echo power of a radar in detecting sandy dust weather. Next,the numerical

calculation of radar echo power propagating through sandy dust weather will be detailed.

### 2.2.1 Meteorological radar detection principle

As shown in Figure 1(a), the radar emits a beam of electromagnetic wave propagating through a sandy dust system, and the relationship between radar receiving power and radar system parameters, detection target characteristics, detection range,attenuation on radar wave back and forth path is described by radar equation. The microwave meteorological radar equation is



(Probert-Jones, 1962; Goldhirsh, 1982)

$$P_r = P_t \frac{G^2 \lambda^2 \Delta\phi \Delta\theta \tau c}{2^{10} \ln 2\pi^2 R^2} \beta \times 10^{-0.2\int_0^R \sigma_{ext} dl} \, , \tag{3}$$

where $P_t$ is the transmitting power and $P_r$ is the receiving power; $R$ is the detection range; $\Delta\varphi$ and $\Delta\theta$ are the beam width; $G$ is the antenna gain; $\lambda$ and $\tau$ are the wavelength and pulse time of the transmitted electromagnetic wave, respectively. In particular, the laser radar (lidar) equation is (Collis, 2010)

$$P_l = P_t \frac{\tau c A}{R^2} \beta \times \exp\left(-2\int_0^R \sigma_{ext} dl\right), \tag{4}$$

in which, $A$ is the area of the aperture of the radar receiving antenna. In formula (3) and (4), $\beta$ and $\sigma_{ext}$ are the backscattering cross section and extinction cross section of sandy dust particles per unit volume, respectively. $\beta$ and $\sigma_{ext}$ can be calculated by the following formula

$$\begin{aligned}\beta &= \int_{r_{min}}^{r_{max}} \pi r^2 Q_{bac} N_0 p(r) dr \\ \sigma_{ext} &= \int_{r_{min}}^{r_{max}} \pi r^2 Q_{ext} N_0 p(r) dr\end{aligned} \, , \tag{5}$$

In which, $r_{min}$ and $r_{max}$ are the maximum and minimum radius of dust particles; $Q_{bac}$ and $Q_{ext}$ are the backscattering coefficient and extinction coefficient by a single dust particle with particle radius $r$, which are dependent on particle properties and related to the electromagnetic wavelength.

**2.2.2 Electromagnetic scattering of a sandy dust particle**

As mentioned above, the particle properties such as particle shape, charges carried by particle affect the $Q_{bac}$ and $Q_{ext}$. Here,
the particles are considered as perfect spheres. The sandy dust particles are always electrified to carry some negative charges(Zhang and Zhou, 2020). As shown in Figure 1(b), when an electromagnetic wave is irradiating on a charged particle, the wave equations are(Bohren and Hunt, 1977)

$$\left.\begin{aligned}\nabla^2 \mathbf{E} + k^2 \mathbf{E} &= 0 \\ \nabla^2 \mathbf{H} + k^2 \mathbf{H} &= 0\end{aligned}\right\} \tag{6}$$

The boundary conditions are as follows(Klačka and Kocifaj, 2007)

$$\left.\begin{aligned}\vec{n} \times (\mathbf{E}_i + \mathbf{E}_s - \mathbf{E}_1) &= 0 \\ \vec{n} \times (\mathbf{H}_i + \mathbf{H}_s - \mathbf{H}_1) &= \sigma_s \mathbf{E}_t\end{aligned}\right\} \tag{7}$$





Here, $\sigma_s$ is surface conductivity of charged particles, it can be given by Drude model, that is, $\sigma_s = \frac{i\eta q_m}{\omega + i\gamma_s}$, $\eta$ represents the particle surface charge density, $q_m = 1.7587 \times 10^{12}$ Ckg$^{-1}$ is the charge-to-mass ratio of a single electron, $\gamma_s = k_B T/\hbar$, $k_B = 1.38 \times 10^{-23}$ J/K and $\hbar = 1.0546 \times 10^{-34}$Js are Boltzmann's constant and Planck's constant, respectively. By expanding the incident field, internal field and scattering field with spherical harmonic vector function, the scattering field and internal field

of sand can be solved(Bohren and Hunt, 1977). Furthermore, the backscattering coefficient and extinction coefficient of sand can be obtained(Bohren and Hunt, 1977)

$$Q_{ext} = \frac{2}{x^2}\sum_{n=1}^{\infty}(2n+1)\operatorname{Re}(a_n + b_n)$$

$$Q_{bac} = \frac{1}{x^2}\left|\sum_{n=1}^{\infty}(2n+1)(-1)^n(a_n - b_n)\right|^2$$

(8)

Here Re($\cdot$) denotes the real part of the complex, and $x$ is the size parameter($x = 2\pi a/\lambda$, where $a$ is the radius of the particle and $\lambda$ is the wavelength of the electromagnetic wave), $a_n$ and $b_n$ are the Mie scattering coefficients of electromagnetic waves

irradiating charged particles(Klačka and Kocifaj, 2007; Zhou and Xie, 2011)

$$a_n = \frac{\psi_n(x)\psi_n'(mx) - m\psi_n'(x)\psi_n(mx) - i\omega k^{-1}\sigma_s \psi_n'(x)\psi_n'(mx)}{\xi_n(x)\psi_n'(mx) - m\xi_n'(x)\psi_n(mx) - i\omega k^{-1}\sigma_s \xi_n'(x)\psi_n'(mx)}\Bigg\}$$

$$b_n = \frac{\psi_n'(x)\psi_n(mx) - m\psi_n(x)\psi_n'(mx) + i\omega k^{-1}\sigma_s \psi_n(x)\psi_n(mx)}{\xi_n'(x)\psi_n(mx) - m\xi_n(x)\psi_n'(mx) + i\omega k^{-1}\sigma_s \xi_n(x)\psi_n(mx)}\Bigg\}$$

(9)

Where $\psi_n(x) = xj_n(x)$ and $\xi_n(x) = xh_n^1(x)$, $j_n(x)$ and $h_n^1(x)$ are first-order Bessel functions and spherical Hankel functions, respectively, $i = \sqrt{-1}$, $\mu_0 = 1.256 \times 10^{-6}$ is vacuum permeability, $m$ is the relative refractive index of dust particles, which is related to the composition of dust particles and the frequency of radar waves. In the process of a sand and dust weather, it

will usually cause climate change in some areas, such as the phenomenon of moisture inversion(Elsheikh et al., 2017). The change of relative humidity will change the moisture content of sand particles. It is found that the moisture content of sand particles also has an effect on the refractive index of sand particles(Ghobrial, 1980). It will further affect the scattering and absorption of electromagnetic waves by sand and dust particles, and eventually affect the echo information of meteorological radar in the detection of sand and dust weather.

Relative refractive index and dielectric constant are important electromagnetic parameters used to describe the interaction between particles and electromagnetic waves, and they can be transformed into each other $m = \sqrt{\varepsilon}$. According to the measured data, Sharif et al fitted the formulas for calculating the real and imaginary parts of the equivalent permittivity of sand particles under different relative humidity(Sharif, 1995; Sami, 2015)





$$\begin{cases} \varepsilon_r'(RH) = \varepsilon' + 0.04RH - 7.78 \times 10^{-4} RH^2 + 5.56 \times 10^{-6} RH^3 \\ \varepsilon_r''(RH) = \varepsilon'' + 0.02RH - 3.71 \times 10^{-4} RH^2 + 2.76 \times 10^{-6} RH^3 \end{cases} \tag{10}$$

In the formula $\varepsilon'$ and $\varepsilon''$ represent the real part and imaginary part of the relative permittivity of dry sand respectively, and the complex permittivity of sand under different relative humidity is $\varepsilon_r(RH) = \varepsilon_r'(RH) + i\varepsilon_r''(RH)$.

### 2.3 Effective detection range of meteorological radar detecting sandy dust weather

Most of meteorological radars used in weather stations are concluded shown as in Table 1. In the meteorological radar detection of sandy dust weather, when the echo power received by the radar receiver is greater than the minimum detectable signal power

(sensitivity), it is considered that the radar system can receive effective echo information, that is,  within the effective detection range of the radar in this band. The sensitivity of the radar system is related to the noise figure and bandwidth of the receiver. The sensitivity and key parameters of radar systems in different bands are listed in Table 1 (Wang et al., 2017; Ming et al., 2019; Oluleye and Ojo, 2020; Chiou and Kiang, 2017). Given radar, the radar echo power depends on the visibility and detection range. Next, the radar echo power is numerical calculated when given the visibility and detection range shown as in

Fig.2, and compared with the radar sensitivity, the white solid lines in Fig.2.

From Figure 2, it can be found that the echo power or receiving power, $P_r$, decreases as the detection range and visibility increasing for L-band radar, S-band radar, X-band radar, Ka-band radar and W-band radar shown as Figs.2(a)-2(e), while it decreases as the visibility increasing and the detection range decreasing for lidar shown as in Fig. 2(f). Given the visibility, it can be obtained a detection range, at which the numerical calculated $P_r$ equal to the radar sensitivity, and we named this

detection range as the effective detection range of the radar. Thus we can obtain the effective detection range of each radar when given the visibility, beyond which,  the radar echo power is too low to detect such sandy dust weather.  In fact, whether the radar system with each band can receive effective echo information or not is mainly related to the backscattering and extinction of radar emitted waves by suspended dust particles. If the radar wavelength is too long (the scale parameter $x = 2\pi a/\lambda$ is too small) or there are few dust particles suspended in the air, the backscattering will be too small, the effective echo power

will be very low as well. In the same way, if the radar wave length is too short (the scale parameter is too large) or the very much dust particles suspend in the air, the radar detection wave attenuates seriously on the back-and-forth path, and the radar system cannot receive effective echo information, either.

(a) L-Band radar (belongs to decimeter band) in detecting sandy dust weather, when the visibility is 10 m, the effective detection range is 2671 m; when the visibility is 10 km, the effective detection range is 178 m. The theoretical research results

of Goldhirsh et al show that the attenuation of L-Band radar wave in low visibility sandstorms is negligible (Goldhirsh, 1982), which means that L-Band radar wave detection of sand and dust weather may not be able to produce effective echo information. However, recent studies by Wang et al.(Wang et al., 2017) show that effective signals can be received when L-Band radar is used to detect four kinds of sandy dust weather. Actually, it is found that whether the sandy dust weather can be effectively



detected is related to the visibility, the detection range and the sensitivity of the radar system. Therefore, the effective detection

range of L-Band radar for detecting sandy dust weather with different strength is given here, to be reference values for the application of this band radar to detect sandy dust weather.

(b, c) S-Band and X-Band radar (centimeter band) in detecting sandy dust weather, the effective detection range is larger than that of 10 km when the visibility is 10 m. When the visibility is 10 km, the effective detection range is 2871 m and 1767 m respectively. Many scholars have studied the transmission characteristics of centimeter wave in sandy dust weather, *e.g.*, the

theoretical research results of Goldhirsh et al. show that extremely severe sandstorms have a significant attenuation of long-range S-Band and X-Band radar waves (Goldhirsh, 1982), while results by Dong et al show that the attenuation of S-Band ($f$=3GHz) and X-Band ($f$=10GHz) electromagnetic waves in general sandstorms is negligible (Dong et al., 2011). Therefore, centimeter-band radar has not yet been used to detect sandy dust weather. Because the centimeter wave is longer than the dust particle size, it may not be able to receive strong backscattering signals. In addition, the centimeter wave radar is distributed

in the fixed stations around the city, although our calculation results show that the centimeter wave radar has a wide range of monitoring, however, for sandstorms that occur in desert areas, it is impossible to detect them from such a long distance. Based on the effective detection range of centimeter wave radar in sandy dust weather of different intensity given in this paper, it can be found that the detection range of centimeter wave radar is very wide. Therefore, we suggest that centimeter wave radar can be used to detect sandstorms around cities. For example, it can be used to detect the recent sandstorm weather in Lanzhou,

Beijing and other places. This is a good supplement to the comprehensive detection of sandy dust weather.

(d, e) Ka-Band and W-Band radar (millimeter band) in detecting sandy dust weather, the effective detection range is larger than that of 10 km when the visibility is 10 m. When the visibility is 10 km, the effective detection range is 1215 m and 8143 m respectively. It is found that the sandy dust weather have a strong attenuation of the transmission of electromagnetic waves in millimeter wave band (Dong et al., 2011). Ming et al. used millimeter wave radar to quantitatively detect the sandy dust

weather in the Taklimadan Desert, and detected three different sand and dust weather processes within the effective detection range of 2000 m (Ming et al., 2019). The effective detection range of the millimeter wave band radar provided by us can provide a reference for the use of millimeter wave band radar to detect sand and dust weather.

(f) Laser radar (visible band) in detecting sandy dust weather, when the visibility is 10 m, the effective detection range is 58m, and when the visibility is 10 km, the effective detection range is larger than 10 km. Laser radar is an active remote sensing

instrument widely used to monitor sandy dust weather, it can detect the vertical distribution of aerosols, but in the sandy dust weather with high concentration, the serious signal attenuation of laser radar occurs.

Actually, there are many meteorological radars are established in weather stations around world. According to the effective detection range of each radar given the visibility, a detection scheme can be established by using the meteorological radars existed in weather station.

**3. The scheme of using meteorological radar to detect sand and dust weather**





In fact, it is important to judge if a radar can detect the sandy dust weather with a visibility. From Fig.3, it can be found that given a weather visibility, the echo power varies with the detection range, and beyond a range (called effective detection range) shown as the dash lines in Fig.3, the echo power is too low to be identified. In Fig.3, the effective detection ranges of radar in each band are calculated when the visibility is 10 km (floating dust weather), 1 km (sandstorm) and 10 m (extremely severe

sandstorm), respectively. The selected relative humidity and particle surface charge density are 80% and -2000 µCm$^{-2}$, respectively. There are obvious differences in backscattering and back-and-forth attenuation of radar waves in different intensity sand and dust weather. As shown in Fig. 3(a), in floating dust weather, the backscattering of low-frequency radar waves by dust particles is very small, so it is unable to produce effective echo information, so the effective detection range is short. For example, the effective detection range of L-Band radar is 210 m. While the backscattering of high-frequency radar

waves is large, and the detection coverage is wide, in fact, the effective detection range of lidar is greater than that of 10 km. As shown in Figure 3b, in the sandstorm weather, the effective detection of decimeter band radar is still the smallest, due to the decrease of visibility, the attenuation of lidar in the back-and-forth path increases, and the effective detection distance begins to decrease, while the microwave band radar due to the enhancement of backscattering, the range of receiving effective echo information becomes larger. As shown in Figure 3c, in the extremely severe sandstorm, the lidar attenuates seriously and

the detection distance is the smallest, which is only 55 m. Low visibility enhances microwave scattering and increases the effective detection range of microwave radar, for example, the effective detection range of L-band radar increases to 3625 m, and the detection range of centimeter wave radar and millimeter band radar exceeds 10 km.

From Fig.3, it can be found that for each weather given visibility, each radar has an effective detection range, beyond which the radar cannot have an effective detection to the sandy dust weather. In reality, the microwave radars are fixed in the weather

station, thus the detection range is limited, and the coverage of sandy dust weather generally does not exceed 10km. When the detection range of the radar is 10km, under which the echo power of the W-band radar, S-band radar and lidar vary with visibility shown as in Fig.4. The minimum detectable signal power of each radar is also shown in Fig.4. Therefore, it can be easily found that the echo power of W-band radar and S-band radar decreases as the visibility increases, while lidar's echo power increases. The lidar wave attenuation is serious and cannot receive the effective echo information in the sandstorm and

the W-Band radar cannot produce the effective echo power in the floating dust weather with high visibility. Combining W-band radar with lidar, the sandy-dust weather with all visibility can be identified shown as in Fig.4(a). If S-band radar and lidar are used together, there is an unidentified visibility zone, about 600-1100m shown as in Fig.4(b). Therefore, in the actual detection of sandy dust weather in desert areas and around cities in medium and small scales, the simultaneous detection of two kinds of radar can comprehensively and accurately detect all the intensity of sandy weather in the whole area. We vividly

draw a schematic diagram of detecting the sand and dust weather, as shown in Fig. 5.

## 4. Influence of grains' charges and relative humidity on scheme

In the sand and dust environment, due to the collision and friction of the dust particles, the dust particles carry a large amount


of net charge(Zheng et al., 2003). Recently, it has been found that the charge carried on the surface of sand and dust particles can reach thousands μCm$^{-2}$(Zhang and Zhou, 2020). Previous studies have shown that the charge carried by sand and dust

particles will affect the extinction and backscattering of electromagnetic waves(Qin and Li, 2015; Xie et al., 2020; Kocifaj et al., 2021). Therefore, it may affect the effective detection range and echo power of radar when detecting sandy dust weather. Below, we study the influence of the excess charge carried by particles on the effective detection range of radar in different bands to detect three typical sandy dust weathers (extremely severe sandstorm V=1 m, sandstorm V=1000 m, and floating dust weather V=10000 m), as shown in Figure 6. In order to study the maximum influence of charge carried by particles, a larger

surface charge density of particles is selected here, $\eta$ =-2000 μCm$^{-2}$. The main results are as follows:

(a) Decimeter band radar (L-band), considering the charge carried by dust particles, when detecting severe sandstorms, the effective detection range is increased 954 m; when detecting sandstorm weather, the effective detection range is increased by 200 m; when detecting floating dust weather, the effective detection range is increased by 49 m. In three kinds of sand and dust weather, the charge carried by particles increases the effective detection range of L-band radar.

(b, c) For centimeter band radar (S-band and X-band), when detecting severe sandstorms, the effective detection range is larger than10km whether considering the charge carried by dust particles or not; when detecting sandstorm weather, the effective detection range is increased by 1205 m and 102 m respectively; when detecting floating dust weather, the effective detection range is increased by 452 m and 43 m, the charge carried by particles increases the effective detection range of S-Band and X-band radar.

(d, e) For millimeter wave band radar (Ka-band and W-band), considering the charge carried by dust particles, when detecting severe sandstorms, the effective detection range is reduced by 100 m and 151 m respectively; when detecting sandstorm weather, the effective detection range is increased by 50 m and 0 m respectively; when detecting floating dust weather, the effective detection range is increased by 41 m and 100 m. The effective detection range of Ka-band and W-Band radar decreases in sandstorm weather, while the effective detection range of Ka-band and W-Band radar increases in the other

two kinds of dust weather.

(f) For visible band radar (laser radar), when detecting severe sandstorm, sandstorm and floating dust weather, the effective detection range is the same with and without considering the charge carried by dust particles, and the charge carried by particles has no effect on the effective detection range of laser radar.

Therefore, here we quantitatively give the influence of the charge carried by particles on the effective detection range of

radar in each band. It can be found that the charge carried by sand and dust particles has an impact on the effective detection range of low-frequency radar, but has almost no effect on the effective detection range of high-frequency radar. It is mainly because the charged particles have an obvious effect on electromagnetic scattering under small-size parameters(Klačka and Kocifaj, 2007). Whether the effective detection range of radar increases or decreases due to the charge carried by particles is related to the intensity of sand and dust weather, radar wave frequency and other factors.

In addition to the charge carried by the particles will affect the effective detection range, the relative humidity will also affect the radar detection range. In the field observation, it is found that the transit of sandstorm will cause the phenomenon of



moisture inversion. Accordingly, previous studies have shown that relative humidity has a significant effect on the propagation of electromagnetic waves in sand and dust weather(Dou and Xie, 2017). Here, the influence of relative humidity (*RH*=80%) on the effective detection range of each band radar to detect three kinds of dust weather is quantitatively analyzed, as shown
in Figure 7. The main results are as follows:

(**a**) For decimeter band radar (L-band), considering the influence of relative humidity, when detecting severe sandstorms, the effective detection range is increased by 552 m; when detecting sandstorms, the effective detection range is increased by 49m; when detecting severe sandstorms, the effective detection range is increased by 31 m. In three kinds of sand and dust weather, the relative humidity increases the effective detection range of L-band radar.

(**b, c**) For centimeter band radar (S-band and X-band), when detecting severe sandstorms, the effective detection range with and without considering the relative humidity is larger than that of 10 km; when detecting sandstorm weather, the effective detection range is increased by 703 m and 301 m respectively; when detecting floating dust weather, the effective detection range is increased by 251 m and 100 m, the relative humidity increases the effective detection range of S-Band and X-band radar.

(**d, e**) For millimeter wave band radar (Ka-band and W-band), considering the influence of relative humidity, when detecting severe sandstorms, the effective detection range is reduced by 502 m and increased by 201 m, respectively; when detecting sandstorm weather, the effective detection range is increased by 703 m and 0m respectively; when detecting floating dust weather, the effective detection range is increased by 252 m and 1154 m, the relative humidity reduces the effective detection range of Ka-band radar in sandstorm weather, and increases the effective detection range of Ka-band and W-Band radar in the
other two kinds of dust weather.

(**f**) For visible band radar (laser radar), when detecting severe sandstorms, the effective detection range is less than 10 m with and without the influence of relative humidity; when detecting sandstorm, the relative humidity increases the effective detection range of laser radar by 602 m; The effective detection range of laser radar considering or not considering the influence of relative humidity is larger than that of 10 km when detecting floating dust weather.

Here we quantitatively give the influence of relative humidity on the effective detection range of each band radar. It can be found that the relative humidity has an influence on the effective detection range of the high frequency radar, but has little influence on the effective detection range of the low frequency radar. The increase or decrease of radar effective detection range caused by relative humidity is related to sandy dust weather intensity, radar wave frequency and other factors.

In the above study, the maximum relative humidity and the charge carried by dust particles are selected, and the maximum
influence of dust particle charge and relative humidity on the effective detection range of radar in each band is analyzed. It is found that in the case of maximum particle charge and relative humidity, although the charge carried by particles and relative humidity will increase or decrease the effective detection range of radar in each band, it has little influence on the selection of appropriate radar band to detect sandy dust weather, and basically does not affect the scheme of using meteorological radar to detect sandy dust weather. However, the relative humidity and the charge carried by particles have a significant impact on the
radar echo power of each band, and the strength of sandy dust weather is inverted from the echo power received, so in order





to obtain the strength of sandy dust weather accurately, we study the influence of both relative humidity and particle charge on the echo power, as shown in Figure 8. We give radar detection range of 100 m and visibility of sand and dust weather of 10000 m. The influence rate of relative humidity and charge carried by particles on echo power is expressed as

$A(RH,\eta) = 100 \times \left( P_r^{RH,\eta} - P_r^0 \right) \Big/ P_r^0$ . It can be found from Figure 8(a, b, c) that the influence rate $A$ increases obviously with

the increase of the surface charge density (horizontal axis) in the low frequency band (L, S, X-BAND), and the maximum influence rate reaches 142% in the low frequency band (L-BAND). However, with the increase of relative humidity (vertical axis), the influence rate $A$ does not change much. It can be found from Figure 8(d, e, f) that in the high frequency band, with the increase of relative humidity (vertical axis), the influence rate $A$ increases obviously from the color change. For example, for laser radar, the maximum influence rate is $A=102\%$, but with the increase of the charge carried by the particles (horizontal

axis), the influence rate $A$ does not change much. The charge carried by particles has a significant effect on the echo power of low-band radar, while the change of relative humidity has a significant effect on high-band radar. Therefore, when retrieving the physical properties of sandstorm based on radar echo power, the influence of humidity should be considered in the high frequency band and the charge carried by particles should be considered in the low frequency band.

## 5. Conclusion

In this paper, firstly, combined with the actual meteorological radar parameters, the effective detection range of each band radar in detecting sandy dust weather with different strength is given. Microwave radar cannot completely detect floating dust weather, but microwave radar has unique advantages in detecting severe sandstorm, because the attenuation of microwave radar in severe sandstorm is smaller than that of high frequency radar. Correspondingly, lidar can detect floating dust weather, but it cannot detect severe sandstorm because of its serious attenuation in severe sandstorm. Based on the effective detection

range of each band radar for sandy dust weather with different strength, this paper presents a scheme for simultaneous detection of sandy dust weather of all strength by combining different band radars. In addition, the effects of relative humidity and particle charge on radar detection of sandy dust weather are studied simultaneously for the first time in this paper. It is found that the charge carried by particles has a significant effect in the low frequency band, and the relative humidity has a significant effect in the high frequency band. We suggest that the effects of humidity and charge carried by particles should be considered

when using meteorological radar to detect sandy dust weather strength information.  However, the effect of particle charge and relative humidity on the effective detection distance is not significant, and it shows these effects will not change the proposed scheme. Finally, pass though this scheme to achieve a comprehensive and accurate detection of all the strength of sand and dust weather in the whole region, release more accurate sandy dust weather strength information, and reduce the harm caused by sandy dust weather.

*Code/Data availability*

The data that support the findings of this study are available from the corresponding author upon reasonable request.



*Acknowledgments*

This research is supported by grants of the National Natural Science Foundations of China (No.11472122) and the Fundamental

Research Funds for the Central Universities (No. lzujbky-2018121). Authors thank Yue Xi for editing the article and helpful
discussions with Dr. Jun Zhou.

*Author contributions*

All authors contributed equally to this manuscript.

*Competing interests*

The authors declare no conflicts of interest.

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

**Table 1 Antenna gain, transmission power and sensitivity of radar systems in each band**

| Frequency | L-band (1.3GHz) | S-band (3GHz) | X-band (9.5GHz) | Ka-band (35GHz) | W-band (100GHz) | Lidar (560THz) |
|---|---|---|---|---|---|---|
| Antenna Gain (dB) | 38.5 | 45 | 41.6 | 40 | 62.6 | / |
| Transmit power (W) | 2360 | 600000 | 50000 | 10 | 2200 | 4000 |
| Sensitivity (dBm) | -112 | -113 | -112 | -106 | -110 | -110 |

# Figures

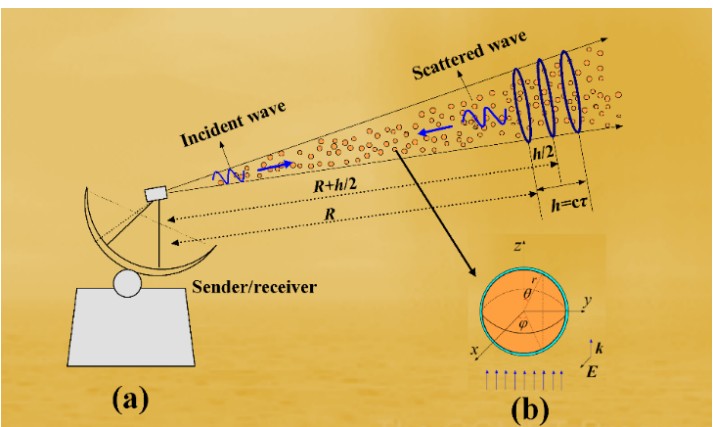

425        **Figure 1. Schematic diagram of meteorological radar detecting sandy dust weather**





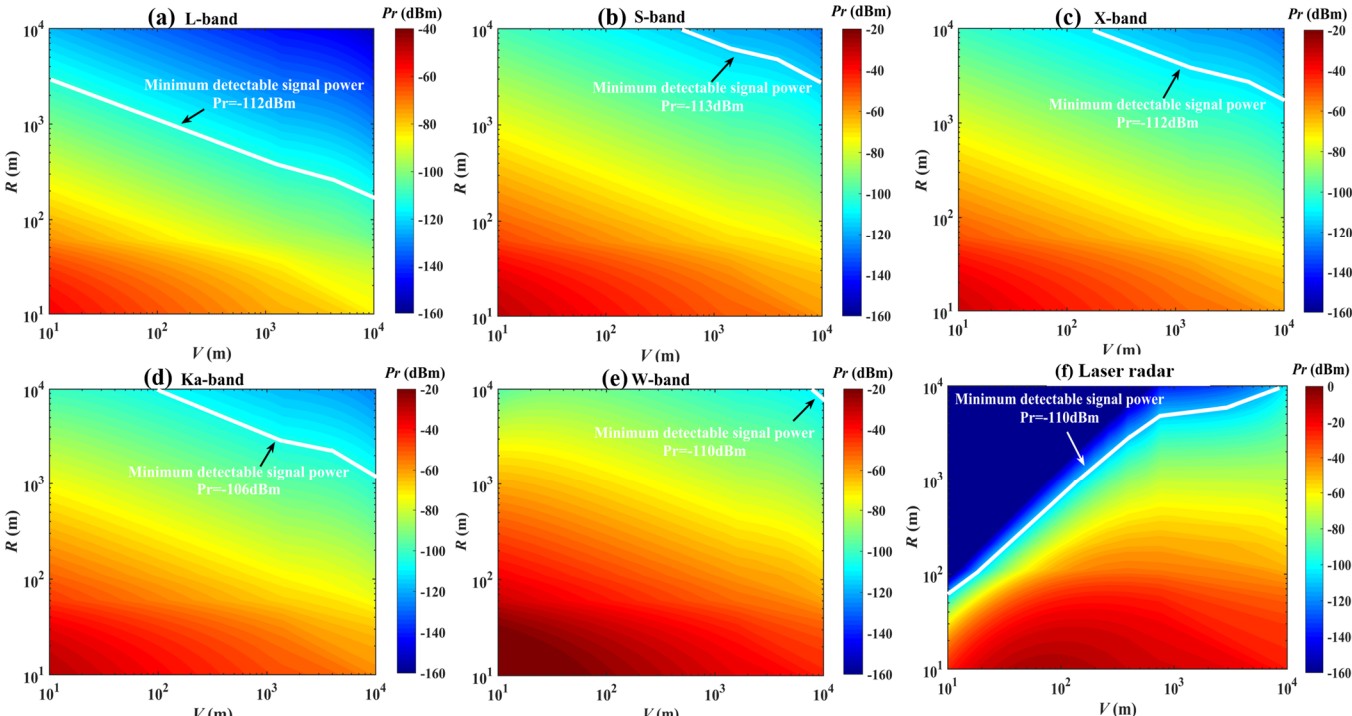

**Figure 2.** Six kinds of radar with different wave bands are used to detect sand and dust weather, the horizontal axis represents the visibility of sand and dust weather, the vertical axis represents the radar detection distance, the color bar represents the echo power received by radar, and the thick white line represents the sensitivity of radar receiver.


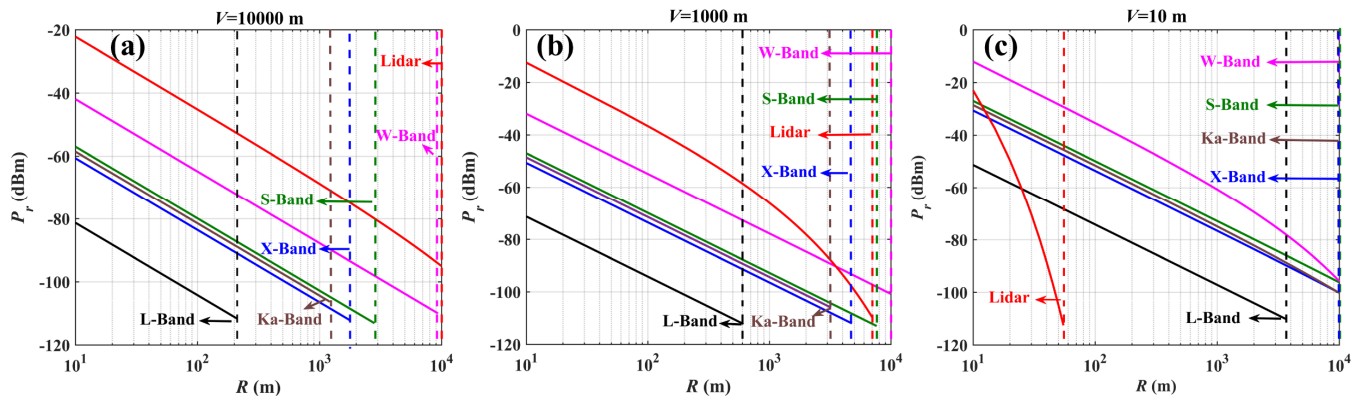

**Figure 3.** The effective detection range of six kinds of radar to detect three kinds of typical sand and dust weather, which characterizes the detection ability of each band radar to three kinds of sand and dust weather.





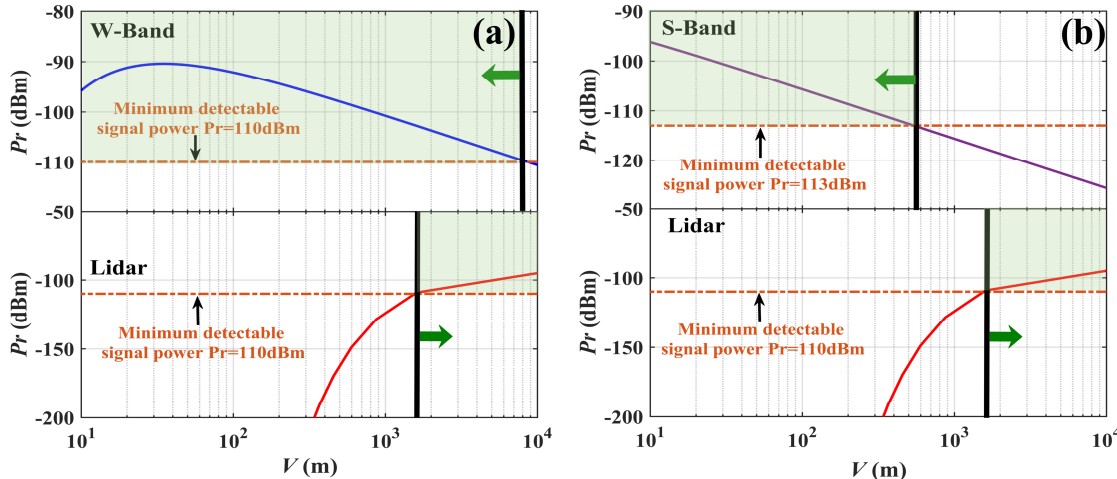

**Figure 4. The radar echo power of the three selected bands varies with the visibility of sand and dust weather, and the green region indicates that the detection distance of effective echo power is 10km, (a) desert area, combined with laser radar and W-band radar to detect sand and dust weather of all intensity, and (b) around the city, combined with laser radar and S-band radar to detect sand and dust weather of all intensity.**

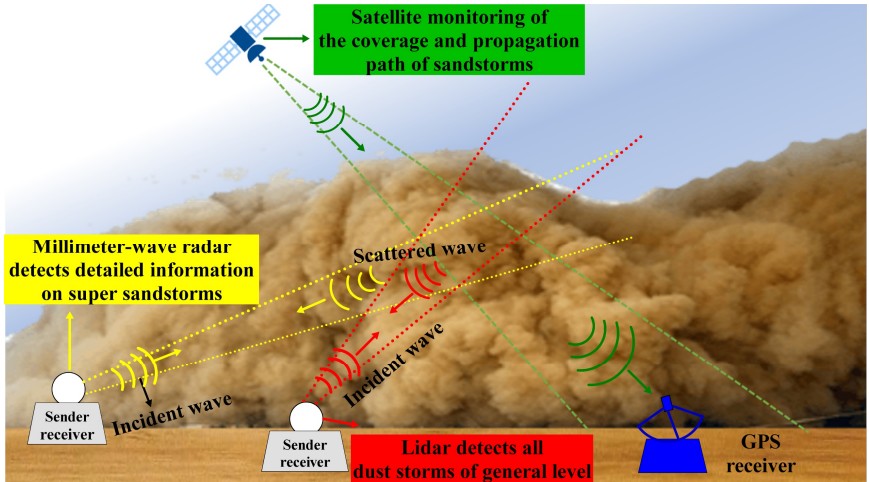

**Fig. 5 A schematic diagram of simultaneous detection of sand and dust weather by lidar and microwave radar, and satellite remote sensing to monitor the coverage and propagation path of sand and dust storms.**





**Figure 6.** The influence of the charge carried by dust particles on the effective detection range of radar in each band under three different sandy dust weathers.





**Figure 7.** The influence of relative humidity on the effective Detection range of Radar in each Band





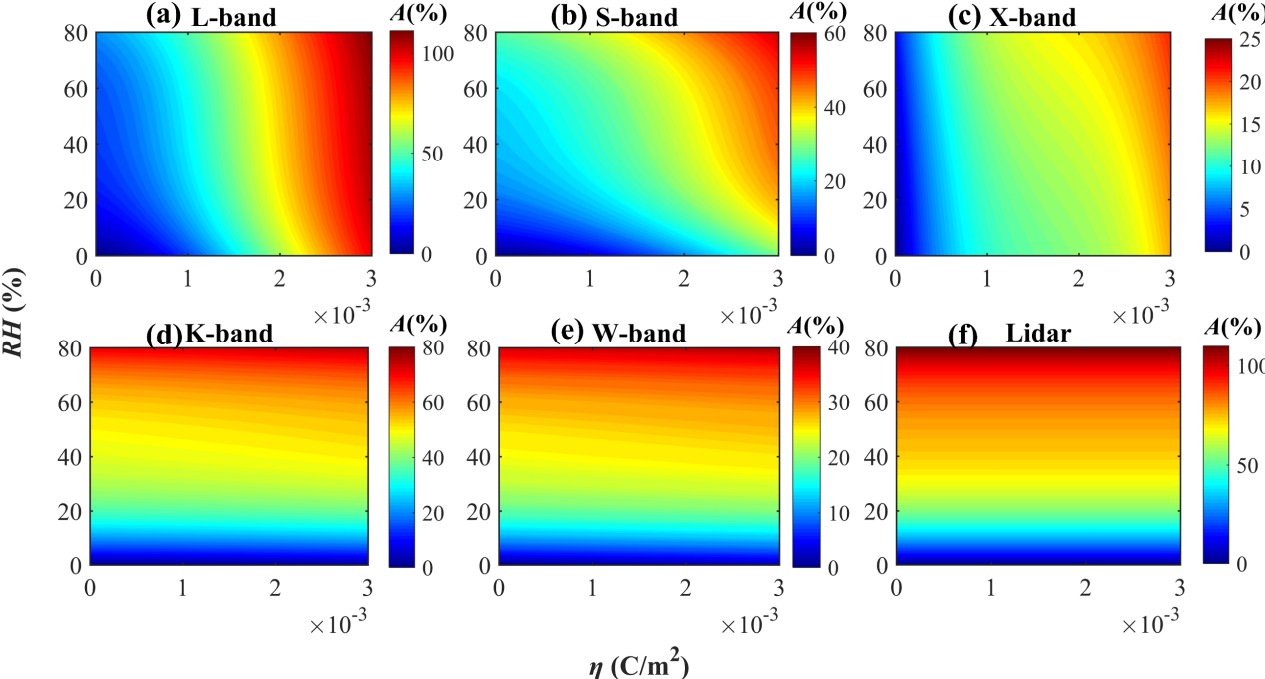

**Figure 8.** At the same time, considering the influence of charge carried by particles and relative humidity on the radar echo power of each band, the color bar indicates the influence rate $A$.