# Peer review of "A scheme to detect sand/dust weather applying meteorological radars"

_Atmospheric Measurement Techniques, 2021_

## Author Comment (AC1)

**Response to reviewers' comments**

Dear Reviewer:

Thank you for your comments on our manuscript. Your comments are valuable for improving our manuscript. We have tried our best to revise the manuscript according to your comments and suggestions, and we have responded to the your comments and suggestions point by point. The following are our answers to editor's and the reviewer's comments.

Thank you very much!

Yours Sincerely,

Li Xie

222 South Tianshui Road,

College of Civil Engineering and Mechanics,

Lanzhou University,

Lanzhou, Gansu 730000, China

Email: xieli@lzu.edu.cn

**Comments from the reviewer:**

Comment 1:

   The paper presents results of simulated detection capability for dust storms by using various radar types and lidar. The simulation considers various dust concentrations mostly expressed as visibility by human eye, and includes effects of relative humidity and electric charge for fixed particle concentrations. Based on the simulation results, the authors propose to use radar of different wavelength and lidar to detect dust storms. The paper does not attempt to calculate the influence of polarization diversity of radar and lidar to distinguish dust storm returns from other returns; this is reasonably not scope of the paper. All calculations are based on spherical particles.

*Authors' reply:* Thanks for your comments. Yes, in this paper we only propose to use radar of different wavelength and lidar to detect the intensity of dust storms, and we do not calculate the influence of polarization of radar and lidar to distinguish dust storm returns from other returns, which is a good suggestion, and we will do it in future. Here, we calculated the detection range of C-band radar by raindrops as following Figure. From the Figure, it can be found easily that detection range is extended compared to the sandy dusty weather, and it can be found that the effective detection range is beyond 100 km except visibility lower than 65 m for raindrops.

[Figure]

Although the sand particles are not always perfect spheres, a study on the shapes of dust particles conducted by Ilan Koren et al. showed that most of dust aerosol particles were spherical (Koren et al., 2001). Therefore, we assume that the sand and dust particles are spherical particles in this paper.

Comment 2:

   The English language is fair to poor which makes it sometimes difficult to follow the authors' argument. The authors should try to seek for support from a native English-speaking person or someone being firm with English language.

*Authors' reply:* We are sorry for the poor descriptions to confuse you. We've checked the manuscript thoroughly and rewritten most parts of the text.

Specific comments:

Comment 3:

Table 1 (and general): Why is C-band radar not considered? It is one of the most

common types of meteorological services.

*Authors' reply:* Thanks for your suggestion, and we include C-band radar in Table 1 and calculate the echo power of C-band radar in Figures.2, 4-6. Results show that it is good choice to detect dusty weather by combining C-band radar and lidar.

Comment 4:

Table 1: Lidar characteristics given indicate a device transmitting visible light with a pulse power of 4 kW. Such lidar is far from being eye-safe and thus not very likely to be used as a scanning device.

*Authors' reply:* We are so sorry that we make a mistake, and the transmit power of lidar (560 THz) should be 110 mJ, and we've corrected it in the revised manuscript.

Comment 5:

Fig.2 (and others): The analysis is limited to a range of 10 km. Most ground-based precipitation radar can detect dust storms at much larger distances; observations beyond 100 km have been reported. The authors should extend the analyses, at least for figures 2 and 3, to at least 50 km (better 100 km) range. Detection ranges below 100 meters need not to be considered.

*Authors' reply:* It is a good suggestion. We calculated echo power of radars and lidar with the maximum detection range 100 km given the visibility as following Figures 1-2 (Figures 5-6 in revised paper). Microwave radars show their excellent ability of detection dusty weather with low visibility, such that C-band radar can get the effective echo power with the detection range of 60~70 km shown as Figure 1(d) and Figure 2(d) when $V$=10 m. According to your suggestion, the echo power of radars and lidar with detection ranges below 100 m was not considered in the revised manuscript.

[Figure]

Figure 1. Echo power varying with the detection range given the visibility (a) 10000 m (floating dust), (b) 1000 m

(blowing sand), (c) 100 m (strong dust storms) and (d) 10 m (severe dust storms), and the solid lines and the dash lines with the consideration of $RH$=80% and $RH$=0%. Particles are neutral. The sensitivity of the six bands of radars/Lidar are -112 dBm, -115 dBm, -112 dBm, -110 dBm, -110 dBm, and -110 dBm, respectively.

[Figure]

Figure 2. Echo power varying with the detection range given the visibility (a) 10000 m (floating dust), (b) 1000 m (blowing sand), (c) 100 m (strong dust storms) and (d) 10 m (severe dust storms), and the solid lines and the dash lines with the consideration of $\eta$=-2000 μCm$^{-2}$ and $\eta$=0 μCm$^{-2}$, without consideration of effect of $RH$. The sensitivity of the six bands of radars/Lidar are -112 dBm, -115 dBm, -112 dBm, -110 dBm, -110 dBm, and -110 dBm, respectively.

Comment 6:
Section 2.3, manuscript lines 172ff: "centimeter-band radar has not yet been used to detect sandy dust weather" and "for sandstorms that occur in desert areas, it is impossible to detect them from such a long distance." Both is not correct. Centimeter wavelength radar is used since decades for dust storm detection, see e.g. Hannesen and Weipert, 2003, and Saeed at al 2014.

*Authors' reply:* We are so sorry that our statements are too arbitrary. We have rewritten these sentences. It is indeed that centimeter wavelength radar is used since decades for dust storm detection as described by Hannesen and Weipert as "Weather Radars are able to detect dust storms, if the number concentration and size of dust particles is large enough(Hannesen and Weipert, 2003)."

Comment 7:
Fig. 2 b and c: According to the authors' calculation, the detection range of centimeter-wavelength radar is limited to about 10 km for visibilities of several hundred meters.

But in Saeed at al 2014 (e.g., figs 6 and 7), detection range of the Kuwait C-band radar is about 100 km for similar visibilities. The authors need to revise their calculation and should comment on such huge discrepancy.

*Authors' reply:* Thanks a lot! We've checked our calculation, and it is correct. We accepted your suggestion, and we calculated the echo power of C-band radar to see Figure 2 in the revised manuscript. The maximum detection range calculated is about 66 km, which is still much lower than the experimental results, about 100km. There are two sides to make such gap between the calculated results and the experimental ones. The one is due to dust particles with different particle sizes in our manuscript and in Saeed et al.'s work, and the other is due to the fact that we assume that the visibility of dusty weather is uniformly distributed along the transmission route, while only partial areas within the detected range appeared dusty weather in Saeed et al(Saeed et al., 2014). Therefore, the attenuation of the radar waves on the transmission path is more severe in our calculations, resulting in a smaller effective detection range than in Saeed et al. We added the comment on the discrepancy between the calculated results and experimental results in the revised manuscript to see lines 56 – 61, page 2. Fortunately, all the echo powers of different radars and lidar are calculated under such same assumption, and it cannot affect to investigate the detection ability of different radars and lidar. Therefore, it cannot affect the scheme. It should further confirm the visibility and the real conditions should be considered in real detection.

Comment 8:
Fig. 2 f and discussion in the text: The authors should compare their derived detection ranges with those according to ISO 28902-2:2017

*Authors' reply:* Thanks for your suggestions. We are sorry that we did not find ISO 28902-2:2017. But fortunately, we find experimental results of detection range of Ka-band radar in dusty weather observed by Ming et al., and we compare it with our calculated results shown as in figure4 (Figure 2(d) in revised manuscript).

[Figure]

Figure 4. Echo power contour maps of Ka-band radar varying with the visibility $V$ of the dusty weather and the detection range $R$, and white dash lines are the minimum detectable signal power (sensitivity) of the given radar or lidar. Squares in (d) are experimental results in dust storms ($V$<1km) at a height of 2000 m and blowing sand (1km<$V$<10km) at a height of 600 m.

Comment 9:
Fig. 3: This figure should be given also for a visibility of 100 meters (not only for 10 and 1,000 meters)
*Authors' reply:* Thanks for your suggestions, and we have calculated the results under visibility of 10m to see Figures 5-6 in the revised manuscript.

Comment 10:
Section 4 (manuscript lines 228-308): The authors describe the influence of electric particle charge and relative humidity in many sentences, with the data given being of limited value. For the reader, e.g., "considering the influence of relative humidity, when detecting severe sandstorms, the effective detection range is reduced by 502 m and increased by 201 m, respectively" means that he has to figure out to which original data such reduction refers to. A reduction by 502 m is significant if it means e.g. from 2,000 down to 1,498 meters, but it is marginal if it means e.g. from 20,000 down to 19,498 meters. Instead of many such sentences, the authors should present a few tables with all these data and should summarize the tables in the text.
*Authors' reply:* We are so sorry that the descriptions make you confused. And we have rewritten this part to see Figures 1-2 above (also to see Figures 5-6 in revised manuscript). It can be found that given $V$, the echo powers of all radars decrease as the detection range increasing, and the $RH$ and charges cannot change this trend of $P_r \sim R$, but $RH$ and charges can enhance the effective detection range such that the effective detection range of lidar with $V$=10000m is enhanced about 5km to see Figure 5(a) with $RH$ increased to 80%, which is 1/5 of the effective detection range ($RH$=0%), about 25 km. From Figure 6(d), it can be found that the excess charge of $\eta$=-2000 µCm$^{-2}$ carried by the particles results in the effective detection range enhanced by 1002 m for the L-band radar, which is 38.3% of the effective detection range by the dry neutral sand particles. However, the same excess charge results in the effective detection range enhanced by only 1.3% for C-band radar.

**References**

Hannesen, R. and Weipert, A.: Detection of dust storms with a C-band Doppler radar, 2003.

Koren, I., Ganor, E., and Joseph, J. H.: On the relation between size and shape of desert dust aerosol, Journal of Geophysical Research: Atmospheres, 106, 18047-18054, 2001.

Saeed, T., Al-Dashti, H., and Spyrou, C.: Aerosol's optical and physical characteristics and direct radiative forcing during a shamal dust storm, a case study, Atmospheric Chemistry and Physics, 14, 3751-3769, 2014.

---

## Author Comment (AC2)

**Response to reviewer' comments**

Dear Editors and Reviewers:

Thank you for your comments on our manuscript. Your comments are valuable for improving our manuscript. We have tried our best to revise the manuscript according to your comments and suggestions, and we have responded to your comments and suggestions point by point as following.

Thank you very much!

Yours Sincerely,

Li Xie

222 South Tianshui Road,

College of Civil Engineering and Mechanics,

Lanzhou University,

Lanzhou, Gansu 730000, China

Email: xieli@lzu.edu.cn

**Comments from the reviewer:**

It is challenging to follow the manuscript due to poor use of English language. I would suggest the authors to have their manuscript checked by a native English speaker or somebody with fine English proficiency prior to the initial submission. The manuscript in its current form strains the voluntary review process.

*Authors' reply*: Thank you very much for your review and suggestions. We are so sorry for the poor descriptions to confuse you. We've checked the manuscript thoroughly and rewritten the text.

Specific comments

1. Please give more details on the chosen parameters for the particle size distributions in section 2.1. How much does the maximum detection range depend on the number of very small particles in relation to fewer larger particles? How are \bar{r} and \sigma_r related to V?

   *Authors' reply*: Generally, particles with a diameter greater than 80 microns are difficult to be directly blown to higher than 2m by wind, while smaller dust particles can rise to a height of several kilometers by strong wind, and can be transported to a long distance. Therefore, the sandy dust particles less than 80 microns are chosen and the particle size distribution obeys lognormal distribution as follows

$$p(r) = \frac{1}{\sqrt{2\pi} \ln \sigma_r} \exp\left[ -\frac{\ln r - \ln \bar{r}}{2(\ln \sigma_r)^2} \right]$$

Where $\bar{r}$ and $\sigma_r$ are the average and standard deviation of the particle radius, respectively. The average particle radius $\bar{r}$ is also a function of height, which can be represented as $\bar{r} = \bar{r}_0 h^{-\gamma_m}$, where $\bar{r}_0$ is the average particle radius at 1 m above ground, and $\gamma_m = 0.15$, $\bar{r}_0 = 18.4$ μm and $\sigma_r = 2$. We have added the information of particle size has been added in section 2.1, please see highlighted lines 81 – 83, page 3.

   The detection range depends on the echo power, which is related to the extinction/backscattering cross section resulting from the particle size distribution. Maximum particle size has big extinction/backscattering cross section, but because the probability of big particle is low, so the echo power due to fewer big particle could be not high. $V = \frac{5.5 \times 10^{-4}}{N_0 a_e^2}$ will decrease as the mean particle size increases.

2. Based on your simulations you should also discuss whether special specifications could be proposed for different radar wavelengths to be better suited for sand detection. This could in include the possibilities of increasing the radar sensitivity by increasing the pulse lengths or integration time. Can spatial resolution be exchanged with sensitivity?

   *Authors' reply*: Thanks for your very valuable suggestions. In this manuscript, the

effective detection ranges of different bands microwave radar and lidar in dusty weather are investigated. And we want to make an appeal to use the radars and lidar at different weather stations combined to detect the dusty weather around cities. Based on the calculated results, it can be found that L-band radar and the C-band radar are suitable to detect server dusty weather like sand storm, while lidar is suitable to detect floating dusty weather. We added a case to calculate the echo power of C-band radar with two transmit powers, and it is found that the higher transmit power can increase the detection rang to see flowing figure (also Figure 7(b) in revised manuscript).

[Figure]

The echo power varying with the visibility V of dusty weather of C-band radar and lidar with two transmit Pt=25 kW and 50 kW, given $R$=10 km. The horizontal dash lines stand for the minimum detectable echo power, sensitivity, of each radar and lidar. Along green arrow, the echo power of radar or lidar is higher than its sensitivity.

3. You should elaborate more on the elements shown in Fig. 5 or leave it out.
*Authors' reply*: We have described in detail the elements shown in Figure 7 in the revised manuscript.

4. In Section 4 it would aid the understanding to discuss the increased backscatter and increased attenuation/scattering due to the surface charge and relative humidity. The assumptions on the vertical distribution of humidity remained unclear to me. Furthermore, the water vapor concentration or at least the assumed air temperature should be give. I would have expected an signal attenuation due the water vapor at least in the W-band by a few dB. I might have missed it in an earlier part, but it is unclear to me if the sand storm is horizontally homogeneous over the whole plane or if it just starts a certain range.
*Authors' reply*: According to your suggestion, the effects of excess charge carried by particles and relative humidity on the backscattering and extinction coefficients of microwave radar and lidar waves, as shown in following figures, also to see Figure 3 in the revised manuscript. It can be found that the relative humidity can enhance the backscattering/extinction coefficient. The effect of excess charge on the backscattering of waves with low frequency more obvious, while the effect of relative humidity on the backscattering of waves with high frequency waves more obvious. The attenuation is mainly determined by extinction coefficient. From following figure, it can be found that

the extinction enhancement of W-band wave is not significant with *RH* increasing, therefore a signal attenuation due the water vapor at least in the W-band by a few dB is occurred when the wave goes through sand storm long path. We assume that the visibility of dusty weather is uniformly distributed along the transmission routes. And assuming that the radar site is in a sandy and dusty environment, only the transmission of radar waves in a sandy and dusty environment is considered. In addition, the effect of relative humidity on attenuation and backscattering considers the variation of equivalent dielectric constant of sand and dust particles at different relative humidity.

[Figure]

Effect of excess charge carried by particles and relative humidity on the backscattering coefficient and extinction coefficient, (a) variation of $Q_{sca}^c / Q_{sca}^0$ with surface charge density, (b) variation of $Q_{ext}^c / Q_{ext}^0$ with surface charge density, (c) variation of $Q_{sca}^{RH} / Q_{sca}^0$ with relative humidity, (d) variation of $Q_{ext}^{RH} / Q_{ext}^0$ with relative humidity. Superscripts $c$ and 0 stand for the extinction/backscattering coefficient by charges particle and corresponding neutral particle.

5. Overall, I am missing a comment on the effect of gaseous attenuation for the simulations.

*Authors' reply*: Here we assume the dusty particles are full of detection path, and only the attenuation by dusty particles and water vapor are considered, and other gaseous attenuation is not considered in our calculation. The attenuation by water vapor is considered in the variation of the equivalent dielectric constant of particle with different relative humidity.

6. Besides the effect of particle charge and humidity, the authors should also discuss the following

How does the beam broadening and Earth's curvature affect the detectability of (shallow) dust storms?

*Authors' reply*: The effect of beamwidth and earth curvature on radar echo power has been investigated by Chiou et al. (Chiou and Kiang, 2017). Their results show that considering the effect of beamwidth and earth curvature improves radar detection accuracy, however, from their calculations, the effect of beamwidth and earth curvature on the effective radar detection range is not significant because ignoring earth curvature only affects the accuracy of the detected particle concentration at different altitudes. We will investigate the effect of beam width, Earth's curvature in future in the scheme.

How good is the assumption of spheres for sand particles?

*Authors' reply*: A study on the shapes of dust particles conducted by Ilan Koren et al. showed that most of dust aerosol particles were spherical, especially for small dust particles (Koren et al., 2001). Here the particles are small with mean radius 18.4um, therefore we calculate the particles as spheres.

Minor comments
L 45: Check if Elsheikh et al. (2017) is the correct reference for moisture inversions in sand storms.

*Authors' reply*: We reconfirmed the cited literature. Elsheikh et al. (2017) is the correct reference for moisture inversions in sand storms. Elsheikh et al. point out that the *RH* increased drastically from approximately 20% to 70% during the dust storm measurement.

L 64: "Meteorological radars are usually used to detect the sandy dust weather". This sentence should be reconsidered. At least in my field of work, meteorological radars are primarily used to observe hydrometeors.
From my understanding, the term "radar" stands for "radio detection and ranging" and is therefore different to a "light detection and ranging" system. Thus, I am confused by the term "lidar radar". Instead, I would personally prefer the simple term "lidar".

*Authors' reply*: We checked full text, and corrected the improper description and statement.

Fig. 1: As the yellow background does add nothing to the understanding of the figure, I would make it white.
Figures 2 and 8 should use one color bar each for all six panels. This makes the panels more comprehensible.

*Authors' reply*: According to reviewer's suggestion, revised the figures to remove the yellow background of Figure 1 and use the one panels of Figure2 and Figure4. It looks much better.

[Figure]

Figure 1. Schematic diagram of meteorological radar detecting sandy dust weather

[Figure]

Figure 2

Figure 4

**References**

Chiou, M. M. and Kiang, J. F.: PWE-based Radar Equation to Predict Backscattering of Millimeter-Wave in a Sand-and-Dust Storm, IEEE Transactions on Antennas & Propagation, PP, 1-1, 2017.

Koren, I., Ganor, E., and Joseph, J. H.: On the relation between size and shape of desert dust

aerosol, Journal of Geophysical Research: Atmospheres, 106, 18047-18054, 2001.

---

## Author Comment (AC3)

**Response to reviewer' comments**

Dear Editors and Reviewers:

Thank you for your comments on our manuscript. Your comments are valuable for improving our manuscript. We have tried our best to revise the manuscript according to your comments and suggestions, and we have responded to your comments and suggestions point by point. The following are our answers to your comments. Editor's and reviewers' comments are in black, and the authors' replies are in blue.

Thank you very much!

Yours Sincerely,

Li Xie

222 South Tianshui Road,

College of Civil Engineering and Mechanics,

Lanzhou University,

Lanzhou, Gansu 730000, China

Email: xieli@lzu.edu.cn

**Comments from reviewer:**

Comment 1:
This manuscript presents theoretical calculations of radar sensitivity to dust particles for radar operating frequencies from L- to W-band. Radar detection thresholds are estimated as a function of range and dust storm intensity quantified by a visibility index. It appears that detecting sand and dust storms with currently deployed weather radars would be a good use of those radars to help protect cities and urban environments.

*Authors' reply*: Thanks for your nice comments. Yes, we hope that a scheme can be established by using radars and lidar in the weather stations around cities to detect the dusty weather to protect cities and urban environment.

Comment 2:
While the manuscript presents theoretical calculations, the manuscript does not show any radar observations of sand or dust that validate the theoretical calculations. Also, contrary to the manuscript title, the manuscript does not present a method or 'scheme' to detect sand or dust with weather radars that discriminates sand or dust radar measurements from backscattered energy from raindrops.

*Authors' reply*: We are so sorry that it makes confused because of our poor English description and statements. In the revised manuscript, we identified the title more clearly as "A scheme to detect the intensity of dusty weather by applying microwave radars and lidar". In this paper, the effective detection ranges of microwave radar and lidar to detect the intensity of dusty weather are investigated in view of the current shortcomings in detecting the intensity of dusty weather. Because the microwave radars are suitable to detect the intensity of severe dusty weather like sand storm, while lidar is suitable to detect the intensity of floating dust weather, it is proposed to detect the intensity of all kinds of dusty weather by using radars and lidar together, and that is the scheme. Yes, it is a good suggestion to discriminate sand and raindrop by backscattering energy or coefficient. It is a pity that we have no enough information about the radars and lidars at weather stations, so we cannot conduct out experimental study to validate the scheme, and here just a theoretical scheme of microwave radar and lidar to detect the intensity of all kinds of dusty weather is proposed. Hope all radars, lidars and even the meteorological satellites can be connected to detect the dusty weather such as sand storms. Fortunately, we find some experimental results of detection range, which can validate our calculated results shown as black squares in following figure, also to see Figure 2(d) in the revised manuscript.

[Figure]

Comment 3:

The manuscript needs a review for English grammar and word usage.

*Authors' reply*: We have carefully checked the whole manuscript and clarify all typos in expressions and corrected the mistakes in grammar in revised manuscript.

Specific Comments

1. Abstract. The abstract does not present the results of the study. Also, the abstract states (line 10) that 'The scheme can be efficient to detect sandy dust weather…" A scheme is not presented in this manuscript, just radar calculations to determine whether simulated radars have the sensitivity to detect sand or dust populations. Rewrite the abstract to describe the purpose of the study, methods of the study, results from the study, and potential impacts from the study.

*Authors' reply*: Thank you for your patience, and we've rewritten abstract as "Detection of the intensity of the dusty weather is important for weather forecasting. In this paper, the effective detection ranges of microwave radar and lidar in dusty weather of different intensities were theoretically calculated, some of which are validated by comparing with the experimental results. The effects of excess charge carried by dust particles and relative humidity on the echo power and effective detection range are also investigated. Based on the effective detection range of microwave radar and lidar, a scheme of combined microwave radar and lidar to detect the intensity of dusty weather is proposed, by using which it makes a good supplement to the current detecting the intensity of dusty weather. Especially, it will be a cost-saving way by using the existed meteorological radars to establish the detection scheme, which will make the precaution against the disastrous weather promising."

2. The manuscript presents scattering calculations of sand and dust particles to determine range detection curves. But, the study does not repeat the calculations for raindrops which would show whether the simulated radars are capable of detecting raindrops. Do the simulated radars have the same sensitivity as operational weather radars? Can the simulated radars detect raindrops at 100 km, or 200 km? Please extend the calculations to raindrops.

*Authors' reply*: The sensitivity is different for different radars and lidar, which is given in published papers as following table. We are so sorry that it makes confused due to our poor English description and statements. In this paper, we are concerned to detect the intensity of sandy dust weather by radars and lidar. It is a good suggestion to calculate the detection range of radars and lidar, and we calculated the detection range of C-band radar by raindrops as following Figure. From the figure, it can be found easily that detection range is extended compared to the sandy dusty weather, and it can be found that the effective detection range is beyond 100 km except visibility lower than 65 m for raindrops.

[Figure]

3. Section 2.2.2, line 74, and line 118. The maximum sand or dust particle is limited to diameters of 80 microns (line 74). The shortest radar wavelength is about 3 mm from W-band radar. The size parameter (line 118) is given as x = 2pi a /lambda. Using a = 40 microns and lambda = 3 mm, the size parameter is approximately 0.15. This maximum size dust particle is still within the Rayleigh scattering regime for W-band radar wavelengths. The Mie scattering approximations (equations 8 and 9) are superfluous and will revert to the Rayleigh approximation for these small size parameters. Section 2.2.2 is making the calculations more complicated than necessary.

*Authors' reply*: Yes, you are right, by reverting Mie scattering theory to Rayleigh approximation, the calculation will become simple. We consider the contribution of charges carried by the particles on the scattering/extinction/echo power, therefore Rayleigh approximation is not suitable.

4. Line 158. I do not know of a civilian scanning weather radar operating at L-band. Most scanning weather radars have antenna beamwidths no larger than 1 degree. An L-band antenna would have to be large to produce a 1 degree beamwidth. If the authors know of an L-band scanning weather radar, it would be interesting to see details of that radar.

*Authors' reply:* We found a literature of a study on the detection of dusty weather intensity by L-band radar as following table (Wang et al., 2013), and some parameters of L-band radar are used in our calculation.

| Name | Parameter | Name | High-mode parameter | Low-mode parameter |
|------|-----------|------|---------------------|---------------------|
| Radar wavelength | 227 mm | Pulse width | 0.66 μs | 0.33 μs |
| Beam width | 8° | Minimum detection height | 600 m | 50 m |
| Beam number | 5 | Noise coefficient | 2 dB | 2 dB |
| Antenna gain | 25 dB | Height resolution | 100 m | 50 m |
| Feeder loss | 2 dB | Coherent accumulation number | 64 | 100 |
| Receiver | Digital IF | FFT points | 512 | 256 |
| Transmitting peak power | 2.36 kW | Bandwidth | 1.5 MHz | 3.0 MHz |

5. Lines 118 to 194. The manuscript presents effective detection ranges with 1-meter resolution. For example, line 159, the detection range is 2671 m. Given the assumptions in the calculations, this is a false sense of accuracy. What are the simulation uncertainties for detection range? Asked another way, given a 3 dB uncertainty in signal-to-noise ratio, what is the uncertainty of the detection range?

*Authors' reply*: We determined the effective detection range by the sensitivity of radar and Lidar. When the calculated echo power of radars or lidar is equal to the sensitivity at a range, defined as the effective detection range. It is inevitable there is calculated error or their noise in the environment, and they will make the effective detection range shorten. But it is theoretical calculation in our manuscript, so we did not consider the uncertainty due to the errors and noise. We hope a detailed analysis to be done.

6. Figure 2. Why do the detection ranges only go out to 10 km when weather radars typically have ranges out to 100 to 300 km?

*Authors' reply:* We have extended the detection distance to 100 km in the revised manuscript as following figures, also to see Figures 5-6 in the revised manuscript. we assume the visibility is uniform along the wave transmit path.The detection range in our calcualtion means the distance of tranmisting path full of dusty particles not the distance between the transmit end and the dusty weather. Therefore, the attenuation of the radar waves on the transmission path is more severe in our calculations, resulting in a smaller effective detection range than the typical radar detection range.

[Figure]

[Figure]

7. Section 2 presented theoretical calculations of radar detection. Are there any radar observations of sand or dust storms that can validate these calculations? Without showing any real radar observations, the simulations have not been validated or put into real-life context.

*Authors' reply:* In this paper, we are concerned to detect the intensity of sandy dust weather by radars and lidar. Because the microwave radars are suitable to detect severe dusty weather like sand storm, while lidar is suitable to detect floating dust weather, it is proposed to detect all kinds of dusty weather by using radars and lidar together. Fortunately, we find some experimental results of detection range, which can validate our calculated results shown as black squares in following figure, also to see Figure 2(d) in the revised manuscript.

[Figure]

8. Section 3 "The scheme of using meteorological radar to detect sand and dust weather". This section does not present a "scheme" or method of detecting sand or dust weather. It appears that some thresholds have been set and shown in Fig. 4, but no flow diagram showing the decision logic is presented in the manuscript. Also, it does not present a method to discriminate scattering from sand or dust from scattering from raindrops. How does the method determine whether sand or dust is being detected rather than raindrops?

*Authors' reply:* We are so sorry that it makes confused because of our poor English description and statements. In this paper, we are concerned to detect the intensity of sandy dust weather by radars and lidar. Because the microwave radars are suitable to detect severe dusty weather like sand storm, while lidar is suitable to detect floating dust weather like W-band/C-band radar and lidar, it is proposed to detect all kinds of

dusty weather by using radars and lidar together, and that is the scheme. Yes, it is a good suggestion to discriminate sand and raindrop by backscattering energy or coefficient. It is a pity that we have no enough information about the radars and lidars at weather stations, so we cannot conduct out experimental study to validate the scheme, and here just a theoretical scheme is proposed. Hope all radars, lidars and even the meteorological satellites can be connected to detect the dusty weather such as sand storms. Fortunately, we find some experimental results of detection range, which can validate our calculated results shown as black squares in following figure, also to see Figure 2(d) in the revised manuscript.

**References**

Wang, M., Wei, W., Ruan, Z., He, Q., and Ge, R.: Application of wind-profiling radar data to the analysis of dust weather in the Taklimakan Desert, Environmental Monitoring and Assessment, 185, 4819-4834, 10.1007/s10661-012-2906-4, 2013.